# Hierarchical Divide-and-conquer Grouping for Generalized Zero-Shot Learning

## Abstract

Generalized Zero-Shot Learning (GZSL) faces a key challenge in transferring knowledge from base classes to classify samples from both base and novel classes. This transfer learning paradigm inherently risks a prediction bias, wherein test samples are disproportionately classified towards the base classes due to the models' familiarity and overfitting to those classes during training. To tackle the prediction bias issue, we introduce a divide-and-conquer strategy that segregates the united label space into distinct base and novel subspaces. Within each subspace, we train a customized model to ensure specialized learning tailored to the distinct characteristics of the respective classes. To compensate for the absence of novel classes, we propose utilizing off-the-shelf diffusion-based generative models, conditioned on class-level descriptions crafted by Large Language Models (LLMs), to synthesize diverse visual samples representing the novel classes. To further relieve the class confusion in each subspace, we propose to further divide each subspace into two smaller subspaces, where the classes in each smaller subspace are obtained with the unsupervised cluster strategy in the text embedding space. With our hierarchical divide-and-conquer approach, the test samples are first divided into a smaller subspace and then predicted the class labels with the specialized model trained with the classes present within the subspace. Comprehensive evaluations across three GZSL benchmarks underscore the effectiveness of our method, demonstrating its ability to perform competitively and outperform existing approaches.

## 1 Introduction

Recently, visual language models (VLMs) such as CLIP Radford et al. (2021) and ALIGN Jia et al. (2021) have significantly facilitated the transfer of knowledge across different tasks or domains. To further adapt the pre-trained VLMs to specific downstream tasks, some researchers propose to fine-tune the models through subtle designed extra learnable layers Gao et al. (2024) or a series of learnable prompts Zhou et al. (2022b) Cao et al. (2024) with some training samples that related to the target tasks or domains. This adaptation hypothesizes that all test samples have the same distribution and are located in the same space as the training samples.

However, when the models are fine-tuned using samples that originate from different classes or domains that the test samples are intended to classify, there may be a domain shift or negative transfer. For example, in Generalized Zero-Shot Learning (GZSL) Xian et al. (2018); Liu et al. (2023), the models are tasked with classifying samples from both base classes, which they were trained on, and novel classes, which they have not seen during training. By fine-tuning the models with only based class samples, the models generally tend to predict all test samples into the same classes as the training samples, resulting in serious prediction bias Wang et al. (2023b).

We argue that the prediction bias arises inherently from the transfer learning paradigm, which encounters test data distributions that deviate significantly from those encountered during training. As shown in Fig. 1(a), in the GZSL setting, if the novel and the base classes are classified in the same test space, it is hard to classify them due to the serious prediction bias. In contrast, the label prediction becomes easier when the test samples are classified into base or novel subspaces. To address the prediction bias issue in GZSL, we propose a divide-and-conquer paradigm for GZSL that divides the joint label space into sub-spaces and trains a separate model for each label space.

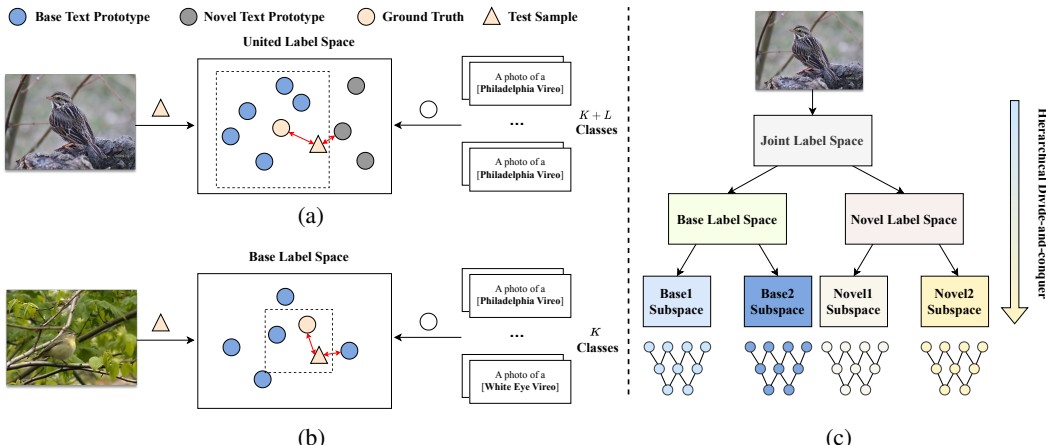

Figure 1: (a) The unified test space is divided into two subspaces, a base label subspace, and a novel label subspace, to alleviate the prediction bias issue. (b) The base label subspace is further divided into two smaller subspaces to mitigate the class confusion issue. (c) The proposed hierarchical divide-and-conquer structure.

Specifically, we introduce a hierarchical divide-and-conquer grouping (HDG) approach, tailored for GZSL. We divide the joint label space into base and novel subspaces and then train a separate model for each subspace. In the absence of visual samples of novel classes, we adopt the off-the-shelf diffusion-based generative models Podell et al. (2023) conditioned on the class-level descriptions generated with GPT-4 Achiam et al. (2023) to synthesize diverse visual samples for each novel class. To encourage the synthesized visual samples to be discriminative, we filter out the samples of inferior quality with a semantic-level concept selection strategy. Given an inference image, we propose a multi-modal distance metric to predict the test images into the correct subspace, where the optimal threshold is determined with a Normalized Mutual Information (NMI) strategy.

The divide-and-conquer paradigm, when applied to base and novel classes, could reduce prediction bias across domains, but it still faces the challenge of class confusion within each subspace, as shown in Fig. 1(b). To this end, we further apply the divide-and-conquer paradigm to each subspace to continue grouping "hypothetical base" and "hypothetical novel" into smaller but focused subsets. Specifically, we adopt the Kmeans method Xie et al. (2016) in the text embedding space to group the classes in each subspace into two clusters, where the classes in each cluster span a smaller subspace. To this end, our method is formulated as a tree structure, as shown in Fig. 1(c).

For each subspace, we train a specific model with the visual samples from the corresponding classes. Specifically, we augment the pre-trained and frozen CLIP visual encoder with an additional trainable linear layer at its output. Note that the training samples for novel classes are synthetically generated. During the inference phase, we first assign test samples to their respective subspaces and then classify the test samples with the specific model.

In summary, our main contributions are:

1. We introduce an innovative hierarchical divide-and-conquer grouping (HDG) paradigm to overcome the prediction issue inherent in traditional transfer learning approaches in GZSL. By employing a hierarchical division of the learning task and subsequently tackling it in a conquering manner, the HDG offers an effective approach to tackling the prediction bias and class confusion that arise in GZSL.

2. We propose to generate diverse visual images conditioned on the class-level descriptions derived from the LLMs and subsequently filter out the low-quality samples with a semantic-level concept selection strategy, which compensates for the scarcity of novel class samples. These diverse synthesized samples ensure the effectiveness of the data grouping and model fine-tuning.

3. Comprehensive experimental evaluations consistently demonstrate that our proposed HDG paradigm achieves substantial improvements over the current state-of-the-art methods across three popular GZSL benchmarks.

## 2 RELATED WORKS

**Generalized Zero-Shot Learning.** GZSL aims to recognize both base and novel classes relying solely on the models trained with the visual samples from base classes. The existing methods can generally be grouped into two lines: (1) Embedding-based methods Naeem et al. (2023); Chen et al. (2022a; 2023) usually align the visual-semantic relationships to connect two different modalities. Once trained, the models can be transferred from base classes to novel ones. However, these methods are constrained to prediction bias towards base classes due to the lack of novel visual data. (2) Generative-based methods Kong et al.; Chen et al. (2021); Xu et al. (2022a) aim to generate batches of visual samples or visual prototypes with semantic descriptions (*e.g.,* sentences, hand-crafted attributes) for novel classes, enabling the model to be formulated by supervised classification fashion. Despite the encouraging advancements that have been made, generative-based methods still grapple with prediction bias, stemming from the essence of transfer learning.

Unlike existing methods, we propose addressing prediction bias by partitioning the test space into distinct subspaces. Our work is most related to calibration-based methods. COSMO Atzmon & Chechik (2019) employs a soft combination strategy to adjust prediction probabilities, while Gating-AE Kwon & Al Regib (2022) uses a binary classification approach to distinguish between base and novel concepts. However, these methods primarily concentrate on the binary classification between base and novel concepts, neglecting further data separation within each domain. In contrast, our approach hierarchically divides the joint class space into subspaces and trains a dedicated model for each subspace.

**Vision-Language Models for Zero-Shot Generalization.** The application of Vision-Language Models (VLMs) such as CLIP Radford et al. (2021) to downstream visual tasks has become increasingly prevalent, thanks to their extensive self-supervised pre-training on vast amounts of web-scale image-text data. By comparing the distance between images and texts, VLMs can achieve open vocabulary recognition using both visual and textual encoders. The emergence of efficient tuning techniques, such as prompt-learning Zhou et al. (2022a); Khattak et al. (2023) and adapter-tuning Gao et al. (2024); Zhang et al. (2022), has further revolutionized this landscape by integrating natural language processing techniques into computer vision. Specifically, CoOP Zhou et al. (2022b) uses learnable prompts to replace hand-crafted templates, while CLIP-Adapter Gao et al. (2024) learns additional Multi-Layer Perceptron (MLP) layers to adapt general VLMs to specific classes. SHIP Wang et al. (2023b) leverages generative models to combine prototype generation with prompt learning. However, these methods are primarily suitable for tasks with consistent data distributions and may not perform well on tasks with inconsistent distributions.

**Data Separation and Grouping.** In the context of open-world classification, the objective of Out-of-Distribution (OOD) detection is to learn a binary classifier that can distinguish between In-Distribution (ID) and OOD samples, a process that is analogous to our grouping approach. For example, MCM Ming et al. (2022) uses the maximum softmax probability derived from CLIP to perform OOD detection. CLIPN Wang et al. (2023a) further improves performance by incorporating negative prompts. Meanwhile, methods such as OOD Chen et al. (2020) and DUS Su et al. (2022) attempt to separate features of novel classes (in the context of GZSL) from base samples using variational autoencoders. However, existing methods primarily focus on the binary separation between ID and OOD distributions. In contrast, we explore the efficacy of a multi-stage hierarchical divide approach that differentiates between various distributions, rather than relying on a single-stage binary classification.

## 3 PROPOSED METHOD

**Problem Formulation.** In GZSL, we distinguish between two distinct and non-overlapping sets of classes: the base classes and the novel classes. The base classes consist of the categories that are present in the training data, while the novel classes are those that are absent during training but are expected to be recognized during testing. GZSL aims to learn a classifier for classifying test samples from both base and novel classes, where the class names of both base and novel classes are available during the training stage.

**Overview of Framework.** The challenge in GZSL lies in reducing the inherent bias towards base classes during prediction, as models are typically fine-tuned exclusively using examples from these

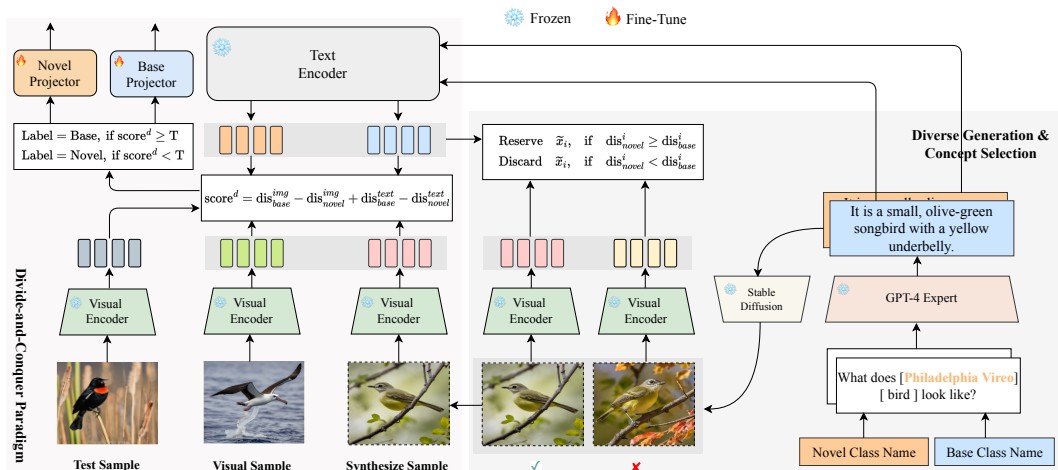

Figure 2: A schematic overview of divide-and-conquer strategy for GZSL. (a) Diverse Generation and Concept Selection aims to generate diversity and semantic-related novel samples with the off-the-shelf stable diffusion models and then filter out semantically irrelevant ones. (b) The Divide-and-Conquer paradigm elegantly partitions the unified label space into distinct base and novel subspaces, employing a multi-model metric as the criterion for division. Within each subspace, specialized models are trained to utilize the classes exclusively belonging to that subspace.

classes. To address this issue, we propose a novel approach that transcends the traditional transfer learning paradigm, which relies solely on models trained on base classes to generalize to both base and novel test samples. Instead, we introduce the Hierarchical Divide-and-Conquer Grouping (HDG) method, which meticulously divides the label space into progressively smaller subspaces. This hierarchical segmentation enables fine-tuning models within distinct subspaces, thereby mitigating prediction bias and label confusion.

As illustrated in Fig. 2, the framework comprises two key components: Diverse Generation and Concept Selection (DGC, detailed in 3.1), and the Divide-and-Conquer Strategy (DCS, described in 3.2). DGC is designed to address the lack of samples for novel classes by synthesizing a diverse range of semantically related samples from the corresponding class names. These synthesized visual samples are subsequently used to fine-tune a dedicated model specifically for the novel classes. On the other hand, DCS divides the unified label space into distinct base and novel label subspaces. It trains separate models for each of these subspaces, leveraging the classes contained within them. To assign test samples to their respective subspaces, a multi-model metric is utilized. During the inference phase, the process involves first classifying each test sample into its corresponding subspace, followed by predictions using the model trained specifically on the classes of that subspace.

## 3.1 DIVERSE GENERATION AND CONCEPT SELECTION

In this section, we leverage readily available generation models to synthesize visual samples for each novel class from the corresponding class name. As shown in Fig. 2, we ask GPT-4 Achiam et al. (2023) to describe the novel classes by the following template:

"Try to describe Class from different poses and scenes as much as possible. Please give Number sentences."

where Class represents the classes name and Number is the number of generated sentences. With the generated descriptions for each novel class, we employ Stable-Diffusion-XL Podell et al. (2023) to synthesize visual samples for the corresponding class. When utilizing GPT-4 to derive class descriptions, as opposed to employing a traditional template like "A photo of Class" Zhu et al. (2024), the resulting samples exhibit greater diversity since the descriptions encompass multifaceted perspectives of each novel class, thus enhancing the range and intricacy of the generated samples.

To provide high-quality samples for the subsequent data processing, we filter out the poorly generated samples based on semantic relevance measurement. Specifically, we calculate the image-to-text (i2t) similarities between all the generated samples and the text descriptions of

both base and novel classes: $\mathrm{dis}_{base}^i = \max\left\{\mathrm{sim}(\mathcal{I}(\widetilde{x}_i), \mathcal{T}(t_y)) \mid \forall t_y \in \mathcal{A}_b\right\}$ and $\mathrm{dis}_{novel}^i = \max\left\{\mathrm{sim}(\mathcal{I}(\widetilde{x}_i), \mathcal{T}(t_y)) \mid \forall t_y \in \mathcal{A}_n\right\}$, where $\widetilde{x}_i \in \mathcal{G}$ is the visual sample from the generated novel data $\mathcal{G}$, $\mathcal{I}$ and $\mathcal{T}$ denote the visual encoder and text encoder of CLIP Radford et al. (2021), respectively. $\mathcal{A}_b$ and $\mathcal{A}_n$ denote the base class set and novel class set, respectively. $\mathrm{sim}(,)$ refers to the cosine similarity.

By comparing the following distances, we pick out samples that are semantically related to their categories with concept selection:

$$\mathcal{G} = \left\{ \begin{array}{llll} \text{Reserve} & \widetilde{x}_i, & \text{if} & \mathrm{dis}_{novel}^i \geq \mathrm{dis}_{base}^i \\ \text{Discard} & \widetilde{x}_i, & \text{if} & \mathrm{dis}_{novel}^i < \mathrm{dis}_{base}^i \end{array} \right. \tag{1}$$

With the concept selection strategy, we selectively retain samples that are semantically pertinent to the novel classes, while disregarding those that are semantically relevant to the base classes. This ensures that the synthesized samples are tightly aligned with the conceptual meanings of the novel classes, fostering a robust representation that is tailored specifically to the novel categories.

## 3.2 DIVIDE BASE AND NOVEL CLASSES AND CONQUER SEPARATELY

The basic idea of our method lies in dividing the label space into smaller sub-spaces and then training an individual model tailored specifically for the classes residing within each subspace. Thus, the key is to correctly classify the test samples into the subspaces. In GZSL, the label space is naturally divided into the base and novel spaces. This section involves extracting the essential multi-modal features to measure differences between base and novel classes.

Specifically, given a test sample $x_j \in \mathcal{X}^{test}$, we formulate the image-to-image (i2i) and image-to-text (i2t) distance varying base to novel classes by:

$$\mathrm{dis}_{base}^{img} = \max\left\{\mathrm{sim}(\mathcal{I}(x_j), \mathcal{I}(x_i)) \mid \forall x_i \in \mathcal{X}_b\right\}, \mathrm{dis}_{novel}^{img} = \max\left\{\mathrm{sim}(\mathcal{I}(x_j), \mathcal{I}(\widetilde{x}_i)) \mid \forall \widetilde{x}_i \in \mathcal{G}\right\},$$
$$\mathrm{dis}_{base}^{text} = \max\left\{\mathrm{sim}(\mathcal{I}(x_j), \mathcal{T}(t_y)) \mid \forall t_y \in \mathcal{A}_b\right\}, \mathrm{dis}_{novel}^{text} = \max\left\{\mathrm{sim}(\mathcal{I}(x_j), \mathcal{T}(t_y)) \mid \forall t_y \in \mathcal{A}_n\right\}, \tag{2}$$

where $\mathrm{dis}_{base}^{img}$ denotes the maximal distance between the image feature of $x_j$ and the image features of the base samples $x_i \in \mathcal{X}_b$, $\mathrm{dis}_{novel}^{img}$ denotes the maximal distance between the image feature of $x_j$ and the image features of generated novel samples $\widetilde{x}_i \in \mathcal{G}$. Similarly, the image-to-text (i2t) distance is calculated with the frozen image and text encoders. $\mathrm{dis}_{base}^{text}$ denotes the maximal distance between the image feature of the test sample $x_j$ and the given base class text features $t_y \in \mathcal{A}_b$, $\mathrm{dis}_{novel}^{text}$ denotes the maximal distance between the image feature of the test sample $x_j$ and the given novel class text features $t_y \in \mathcal{A}_n$. Note that the CLIP encoders are frozen during the training stage.

To separate entangled test samples into independent spaces, we first group the unified test space into independent ones, *i.e.,* "hypothetical base" and "hypothetical novel" by domain score:

$$\mathrm{score}^d = \mathrm{dis}_{base}^{img} - \mathrm{dis}_{novel}^{img} + \mathrm{dis}_{base}^{text} - \mathrm{dis}_{novel}^{text},$$
$$\text{Label} = \text{Base} \quad \text{if} \quad \mathrm{score}^d \geq \mathrm{T}, \tag{3}$$
$$\text{Label} = \text{Novel} \quad \text{if} \quad \mathrm{score}^d < \mathrm{T}.$$

Intuitively, if the test sample belongs to the novel classes, the distance of $\mathrm{dis}_{novel}^{img}$ and $\mathrm{dis}_{novel}^{text}$ would be larger than $\mathrm{dis}_{base}^{img}$ and $\mathrm{dis}_{base}^{text}$, resulting in smaller $\mathrm{score}^d$ than base classes samples, and vice versa. Therefore, we can search for an optimal threshold T to assign the proxy label for each test sample.

To provide a general solution, we suggest utilizing the Normalized Mutual Information (NMI) Estévez et al. (2009) metric as a criterion for determining the optimal threshold, rather than relying on posterior knowledge. This involves first assigning labels of 0 to the base images $\mathcal{X}_{base}$ and 1 to the generated images $\mathcal{G}$. Subsequently, we predict hypothetical classes using Eq. 3. By evaluating the NMI between the ground truth labels and the predicted labels across various threshold values (T), we can quantify the extent of information about the true class labels captured by the specific clustering or grouping outcomes.

### 3.3 HIERARCHICAL GROUPING

Once the whole label space is divided into the base and novel label spaces, the entanglement between classes within both subspaces can significantly impact the final classification outcomes, particularly for fine-grained datasets. To address this issue, we further refine each subspace by dividing it into two additional subspaces, progressively narrowing the class space to mitigate inter-class entanglement.

Specifically, we propose to further divide the "hypothetical base" and "hypothetical novel" into smaller spaces with binary-tree structures. To divide the classes in the base label space into two groups, we apply the Kmeans Kanungo et al. (2002) method to cluster the feature embeddings of text descriptions from the parent classes (*i.e.,* hypothetical base and hypothetical novel) with $K = 2$:

$$
\begin{aligned}
\text{Base1, Base2} &= \text{Kmeans}(t_y, \text{K} = 2), \ t_y \in \mathcal{A}_b, \\
\text{Base1} &\subseteq \mathcal{A}_b, \ \text{Base2} \subseteq \mathcal{A}_b, \ \text{Base1} \cap \text{Base2} = \emptyset,
\end{aligned}
\tag{4}
$$

where Base1 and Base2 are subsets of base classes and K represents the number of clusters. Similarly, we also separate novel classes into two subsets:

$$
\begin{aligned}
\text{Novel1, Novel2} &= \text{Kmeans}(t_y, \text{K} = 2), \ t_y \in \mathcal{A}_n, \\
\text{Novel1} &\subseteq \mathcal{A}_n, \ \text{Novel2} \subseteq \mathcal{A}_n, \ \text{Novel1} \cap \text{Novel2} = \emptyset,
\end{aligned}
\tag{5}
$$

where Novel1 and Novel2 are subsets of novel classes.

Once the anchor classes are selected, we divide "hypothetical base" into Base1 and Base2 based on semantic relevance metric:

$$
\text{score}^b = \text{dis}_{base1}^{text} - \text{dis}_{base2}^{text},
\tag{6}
$$

where the similarities are formulated by $\text{dis}_{base1}^{text} = \max \{ \text{sim}(\mathcal{I}(x_j), \mathcal{T}(t_y)) \mid \forall t_y \in \mathcal{A}_{Base1} \}$, and $\text{dis}_{base2}^{text} = \max \{ \text{sim}(\mathcal{I}(x_j), \mathcal{T}(t_y)) \mid \forall t_y \in \mathcal{A}_{Base2} \}$. Different from the threshold selection of T in the first depth, the deeper class grouping requires rigorous judgment criteria to avoid destroying the original reasonable classification. In this work, we set 0 as the threshold of $\text{score}^b$, in other words, if $\text{score}^b$ is larger than 0, the test sample is assigned to Base1 classes, and vice versa. Similarly, as for "hypothetical novel" classes, we also organize the same deeper grouping process based on $\text{score}^n = \text{dis}_{novel1}^{text} - \text{dis}_{novel2}^{text}$.

### 3.4 FINE-TUNING MODELS FOR GENERALIZED ZERO-SHOT CLASSIFICATION

With the hierarchical grouping strategy, we segregate test samples into multiple independent class sets, which encourages the classification of test samples using models fine-tuned specifically to the distribution from which they originate. For each distinct class set, we customize a model by appending a linear layer to the output of the visual encoder. This linear layer is subsequently trained solely on the samples within the corresponding subset, while the visual encoder remains unchanged during this phase. During the inference stage, test samples are initially categorized into their respective subspaces and then predicted using the fine-tuned models tailored to the classes within those subspaces.

## 4 EXPERIMENTS

### 4.1 EXPERIMENTAL SETUP

**Datasets and Evaluation Metrics.** We evaluate our model across three popular benchmarks. The benchmarks consist of AwA2 Farhadi et al. (2009), a coarse-grained dataset with 50 classes, and two fine-grained datasets: CUB Wah et al. (2011), featuring 200 bird species, and SUN Patterson et al. (2014), with 717 scene categories. Following the GUB protocol provided by Xian et al. (2018), all datasets are divided into two distinct domains, a base domain, which encompasses a set of base classes, each containing a variety of visual samples, and a novel domain, comprising novel classes that are devoid of any visual samples. The performance is evaluated by the harmonic mean of the average per class Top-1 accuracy: $\mathbf{H} = 2 \times \mathbf{B} \times \mathbf{N}/(\mathbf{B} + \mathbf{N})$, where $\mathbf{N}$ denotes the accuracy of novel classes and $\mathbf{B}$ denotes the accuracy of base classes. Notably, while our method divides the original test space into smaller subsets, we refrain from changing the split of test sets to maintain fairness in comparisons.

Table 1: GZSC performance (%) comparisons on three benchmarks. **E**, **G**, and **CA** represent Embedding, Generative, and Calibration-based methods, respectively. The best and second-best results are marked by **bold** and underline. "*" denotes the results reproduced by ourselves.

| Paradigm | Methods | Venue | AwA2 | | | CUB | | | SUN | | |
|---|---|---|---|---|---|---|---|---|---|---|---|
| | | | B | N | H | B | N | H | B | N | H |
| E | MSDN Chen et al. (2022b) | CVPR'2022 | 74.5 | 62.0 | 67.7 | 67.5 | 68.7 | 68.1 | 34.2 | 52.2 | 41.3 |
| | I2MVFomer Naeem et al. (2023) | CVPR'2023 | 79.6 | 75.7 | 77.6 | 59.9 | 42.5 | 49.7 | - | - | - |
| | VGSE Xu et al. (2022b) | CVPR'2022 | 81.8 | 51.2 | 63.0 | 45.5 | 21.9 | 29.5 | 31.8 | 24.1 | 27.4 |
| | CC-ZSL Cheng et al. (2023) | TCSVT'2023 | 83.1 | 62.2 | 71.1 | 73.2 | 66.1 | 69.5 | 36.9 | 44.4 | 40.3 |
| | CLIP Radford et al. (2021) | ICML'2021 | 92.9 | 86.6 | 89.6 | 55.1 | 54.9 | 55.0 | 40.2 | 49.4 | 44.3 |
| | CoOP Zhou et al. (2022b)* | IJCV'2022 | **95.3** | 72.7 | 82.5 | 63.8 | 49.2 | 55.6 | 49.3 | 53.5 | 51.3 |
| | DUET Chen et al. (2023) | AAAI'2023 | 84.7 | 63.7 | 72.7 | 72.8 | 62.9 | 67.5 | 45.8 | 45.7 | 45.8 |
| | PSVMA Liu et al. (2023) | CVPR'2023 | 77.3 | 73.6 | 75.4 | 77.8 | 70.1 | 73.8 | 45.3 | 61.7 | 52.3 |
| | MaPLe Khattak et al. (2023)* | CVPR'2023 | 94.7 | 55.4 | 70.0 | 77.1 | 70.0 | 73.4 | 58.3 | 41.0 | 48.1 |
| | SHIP Wang et al. (2023b) | ICCV'2023 | 94.4 | 84.1 | 89.0 | 58.9 | 55.3 | 57.1 | - | - | - |
| | I2VMFormer Naeem et al. (2023) | CVPR'2023 | 79.6 | 75.7 | 77.6 | 59.9 | 42.5 | 49.7 | - | - | - |
| | ZSLViT Chen et al. (2024) | CVPR'2024 | 84.6 | 66.1 | 74.2 | 78.2 | 69.4 | 73.6 | 48.4 | 45.9 | 47.3 |
| G | Mixup Xu et al. (2022a) | TNNLS'2022 | 69.7 | 60.3 | 64.7 | 60.7 | 58.8 | 59.7 | 38.4 | 46.3 | 42.0 |
| | HSVA Chen et al. (2021) | NIPS'2021 | 76.6 | 59.3 | 66.8 | 58.3 | 52.7 | 55.3 | 39.0 | 48.6 | 43.3 |
| | DENet Ge et al. (2024) | TMM'2024 | 84.8 | 62.6 | 72.0 | 71.9 | 65.0 | 68.3 | 40.8 | 52.3 | 45.8 |
| | ICCE Kong et al. | CVPR'2022 | 82.3 | 65.3 | 72.8 | 65.5 | 67.3 | 66.4 | - | - | - |
| | DFCA Su et al. (2023) | TCSVT'2023 | 81.5 | 66.5 | 73.3 | 63.1 | 70.9 | 66.8 | 40.8 | 52.3 | 45.8 |
| | VADS Hou et al. (2024) | CVPR'2024 | 83.6 | 75.4 | 79.3 | 74.6 | 74.1 | 74.3 | 49.0 | 64.6 | 55.7 |
| | D³GZSL Wanget al. (2024) | AAAI'2024 | 76.7 | 64.6 | 70.1 | 69.1 | 66.7 | 67.8 | - | - | - |
| CA | COSMO Atzmon & Chechik (2019) | CVPR'2019 | - | - | - | 60.5 | 41.0 | 48.9 | 40.2 | 35.3 | 37.6 |
| | OOD Chen et al. (2020) | ECCV'2020 | 75.9 | 55.6 | 64.2 | 50.2 | 49.5 | 49.8 | 33.9 | 41.7 | 37.0 |
| | DUS Su et al. (2022)* | CVPR'2022 | 77.2 | 63.6 | 69.7 | 60.2 | 52.1 | 55.9 | 45.6 | 49.3 | 47.4 |
| | GatingAE Kwon & Al Regib (2022) | TIP'2022 | 81.3 | 60.3 | 69.3 | 58.1 | 55.4 | 56.7 | 38.1 | 45.3 | 41.4 |
| | SZSL Shen et al. (2021) | TCSVT'2021 | 77.5 | 52.8 | 62.8 | 57.7 | 47.6 | 52.2 | 41.7 | 33.5 | 37.1 |
| | HDG-CLIP (Depth=1, Frozen) | **Ours** | 93.0 | 90.2 | 91.6 | 57.1 | 71.5 | 63.5 | 45.7 | 66.1 | 54.0 |
| | HDG-CLIP (Depth=1, Tuning) | **Ours** | 93.9 | 94.2 | 94.0 | 73.8 | 75.2 | 74.5 | 79.8 | 71.2 | 75.3 |
| | HDG-CLIP (Depth=2, Tuning) | **Ours** | 94.5 | **94.1** | **94.3** | **78.4** | **78.0** | **78.2** | **81.4** | 73.3 | **77.1** |
| | HDG-CLIP (Depth=3, Tuning) | **Ours** | 93.5 | 90.3 | 91.9 | 78.4 | 77.5 | 77.9 | 78.5 | 71.4 | 74.8 |

Table 2: Effectiveness analysis of the proposed grouping strategy. with v and w/o v represent with and without image-level distance $\text{dis}^{\text{img}}$ in Eq. 3, respectively. $\text{Acc}_B^C$ and $\text{Acc}_B^F$ represent the accuracy of base classes at depth=1 and depth=2 grouping stages, respectively. $\text{Acc}_N^C$ and $\text{Acc}_N^F$ represent the accuracy of novel classes at depth=1 and depth=2 grouping stages, respectively.

| Methods | AwA2 | | | | CUB | | | | SUN | | | |
|---|---|---|---|---|---|---|---|---|---|---|---|---|
| | $\text{Acc}_B^C$ | $\text{Acc}_N^C$ | $\text{Acc}_B^F$ | $\text{Acc}_N^F$ | $\text{Acc}_B^C$ | $\text{Acc}_N^C$ | $\text{Acc}_B^F$ | $\text{Acc}_N^F$ | $\text{Acc}_B^C$ | $\text{Acc}_N^C$ | $\text{Acc}_B^F$ | $\text{Acc}_N^F$ |
| MLP | 99.8 | 21.5 | 98.3 | 82.6 | 99.8 | 37.8 | 96.4 | 87.3 | 99.3 | 12.6 | 92.5 | 68.4 |
| Xgboost | 91.9 | 77.8 | 97.5 | 88.5 | 97.2 | 94.4 | 95.5 | 90.5 | 90.9 | 68.3 | 85.5 | 81.5 |
| HDG w/o v (Ours) | 96.4 | 97.8 | 99.0 | 98.3 | 94.1 | 92.9 | 97.0 | 95.2 | 91.4 | 92.2 | 94.2 | 93.1 |
| HDG with v (Ours) | 98.3 | 98.8 | 99.2 | 99.6 | 97.8 | 96.4 | 97.2 | 95.5 | 94.6 | 94.1 | 96.0 | 94.5 |

**Implementation Details.** In the DGC module, we leverage GPT-4 Achiam et al. (2023) to generate the descriptions of each class. For image generation, we adopt Stable-Diffusion-XL Podell et al. (2023) to generate 50 samples for each novel class with 10 descriptions provided by GPT-4. In the HG module, we apply the cosine similarity as the distance metric and select the optimal threshold T based on the normalized mutual information (NMI) Estévez et al. (2009). For the deeper divided process, we adopt $\text{Kmeans}$ Kanungo et al. (2002) with $K = 2$ in the text semantic embedding space to separate the leaf node classes.

## 4.2 COMPARISONS WITH SOTAS

Table. 1 presents a comparative analysis of the proposed method against recent GZSC competitors. From the results, it is evident that the proposed HDG-CLIP (Depth=2, Tuning) performs the best in terms of both **N** and **H** metrics across all three datasets, which significantly surpasses the second-best competitors with large margins, achieving improvements of 4.7% on the AwA2 dataset, 3.9% on the CUB dataset, and a substantial 21.4% on the SUN dataset in terms of the **H** metric.

Compared to MaPLe Khattak et al. (2023), which fine-tunes the VLMs with only base classes through prompt learning, HDG-CLIP (Depth=2, Tuning) significantly improves the novel class accuracy (close to 40%) without sacrificing the base class performance. This notable improvement stems from the adoption of a divide-and-conquer paradigm, coupled with tailored fine-tuning mod-

els specifically designed for base and novel classes, respectively. Compared with the original CLIP Radford et al. (2021) that classifies the test samples into the unified label space, our HDG-CLIP (Depth=1, Frozen) obtains better performances across all evaluated metrics on all three datasets, which indicates that the divide-and-conquer strategy relieves the confusion in the unified label space. Additionally, the HDG-CLIP model, when fine-tuned specifically for each segmented label space (Depth=1, Tuning), demonstrates substantial performance gains over its non-fine-tuned counterpart, emphasizing the significant potential of targeted fine-tuning to enhance the overall discriminative capabilities of the method. In contrast to HDG-CLIP(Depth=1, Tuning), HDG-CLIP(Depth=2, Tuning) obtains notable performance improvements on all three datasets in terms of three metrics, particularly evident in fine-grained datasets. Additionally, when the label space is segmented into even more refined subsets and the models are fine-tuned exclusively for these smaller, leaf-level label spaces in the context of HDG-CLIP with a depth of 3, it results in a decrease in performance. This indicates that excessive granularity in the labeling and fine-tuning process may not always lead to improved results, potentially due to overfitting or reduced generalization capabilities.

### 4.3 FURTHER ANALYSIS

**Ablations of Hierarchical Grouping.** In this experiment, we evaluate the effectiveness of the proposed sub-space divided strategy. Specifically, we select two methods for comparison, including a simple classification model achieved with an MLP and Xgboost method Chen & Guestrin (2016). For these two methods, we first label the samples in each depth. For the first depth, we label the training samples and generation samples with 0 and 1. For deeper depth, we label the samples in each subset based on the clusters. Then, we train the models for each subset.

As shown in Table. 2, supervised methods usually achieve superior accuracy on base classes, while sacrificing the $\text{Acc}_\text{N}^\text{C}$. This reflects that domain classifiers obtained through supervised training tend to overfit to base classes due to the unclear intra-class correspondences. In contrast, HDG effectively uncovers the intricate domain relationships by leveraging a comparative analysis of the inherent characteristics present within multi-modal features. This strategy not only mitigates the risk of overfitting but also enhances the model's robustness. Furthermore, we have observed substantial gains, exceeding 1% improvement, in both base and novel class performance, when incorporating visual information into the learning process. These notable enhancements are primarily attributable to the inherent richness and diversity of visual data.

**Impacts of Threshold** $\text{T}$**.** In Fig. 3, we report the distributions of $\text{score}^\text{d}$, $\text{score}^\text{b}$ and $\text{score}^\text{n}$, respectively. Notably, the first column highlights a strikingly low fractional coincidence across different domains, underscoring the remarkable ability of our method to effectively segregate the base and novel domains within a straightforward and training-free framework.

Furthermore, as depicted in Fig. 5(a), we observe fluctuations in the NMI scores in response to variations in the threshold parameter $\text{T}$. Notably, the $\text{argmax(NMI)}$ consistently aligns with the watershed of $\text{score}^\text{d}$ presented in the first column of Fig. 3. This compelling observation underscores the capability of the proposed NMI metric to serve as a reliable guide for selecting the optimal threshold across diverse datasets.

Besides, the posterior watershed separating the fine-grained grouping scores $\text{score}^\text{b}$ and $\text{score}^\text{n}$ is observed to be 0, as evident from the second and third columns of the figure. This signifies that the distance metric formulated in Eq. 6 adeptly captures the intricate relationships between test samples and their respective leaf anchors (such as Base1 and Base2). This finding underscores the validity and soundness of our proposed unsupervised clustering strategy for achieving precise fine-grained grouping.

**Impacts of Generated Samples.**

To evaluate the impact of sample diversity and concept selection during the sample generation process, we conducted a series of experiments presented in Fig. 4(a). The results indicate a significant 6.3% reduction in the metric **H** when descriptions guided by LLMs are absent (denoted as **w/o LLMs**), which highlights the crucial role of the diverse and semantically relevant descriptions provided by LLMs in promoting high-quality and contextually coherent generations. Furthermore, our method demonstrates an additional 1.5% improvement in **H** on the CUB dataset compared to the scenario without concept selection (denoted as **w/o selection**). This notable enhancement validates

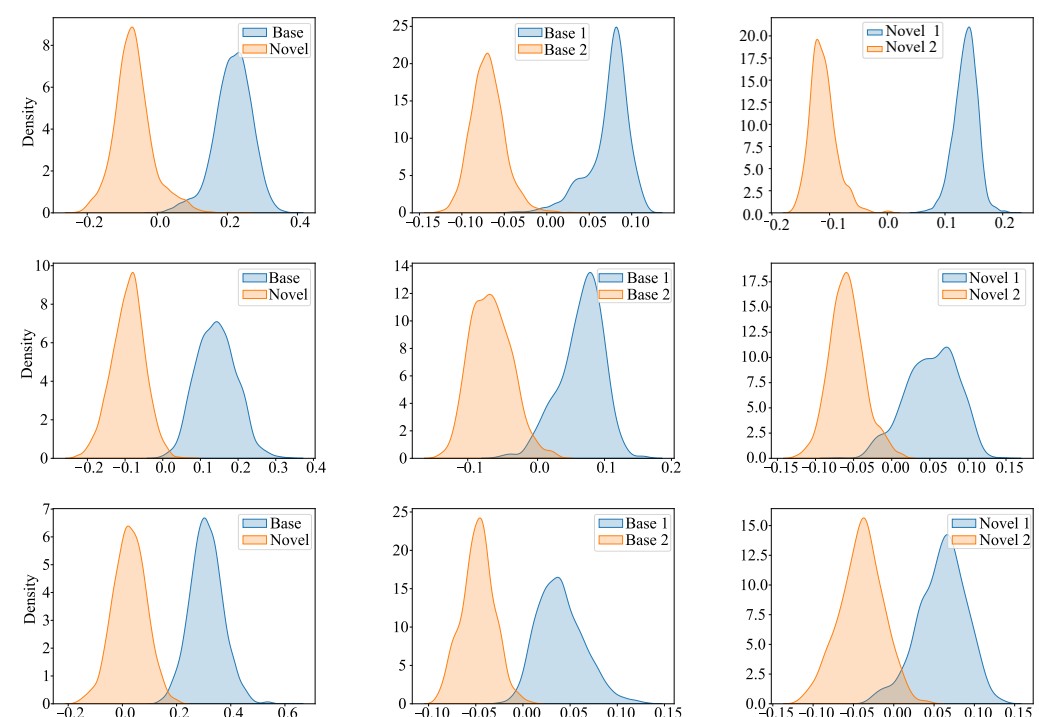

Figure 3: From top to bottom, we report the density maps of $\text{score}^{d}$ (first column), $\text{score}^{b}$ (second column), and $\text{score}^{n}$ (third column) on AwA2, CUB, and SUN, respectively.

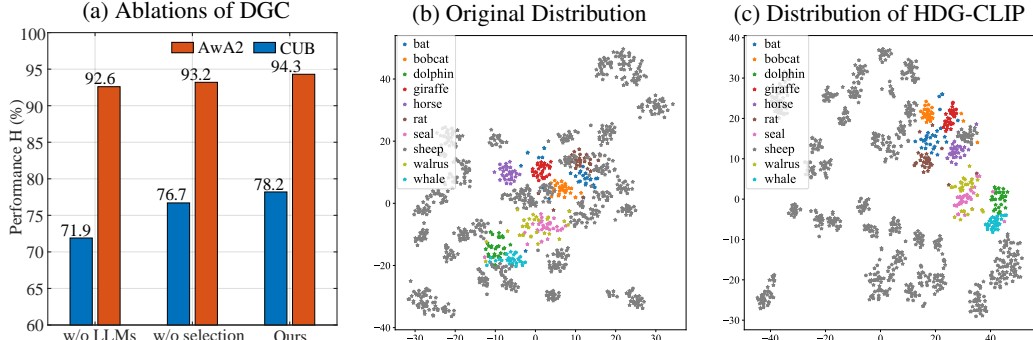

Figure 4: (a) Ablations of DGC. w/o LLMs represents the variant of HDG without LLMs-guided generation and concept selection. w/o selection represents the variant of HDG without concept selection. (b-c) The original distribution of test samples on AwA2 and fine-tuned distribution equipped with HDG-CLIP, respectively. Gray stars represent the base class, and colored stars represent the novel class.

the effectiveness of our concept selection mechanism in removing redundant information from the generated samples, thereby enhancing their quality.

Fig. 5(b) demonstrates the impacts of varying the number of generated samples on the model's performance. Specifically, as the number of samples increases up to 20, a clear upward trend in performance is evident. However, beyond this threshold, the performance fluctuations remain relatively stable, suggesting that there may be diminishing returns and no further significant improvement is achievable by increasing the number of samples.

In addition, we provide visualizations of the attention maps for both real images and generated images in Fig. 5(c). These visualizations reveal that the generated images exhibit similar attention patterns to the real images, thereby confirming the efficacy of the LLMs-guided descriptions and the diffusion-based generator in producing realistic and attention-consistent imagery.

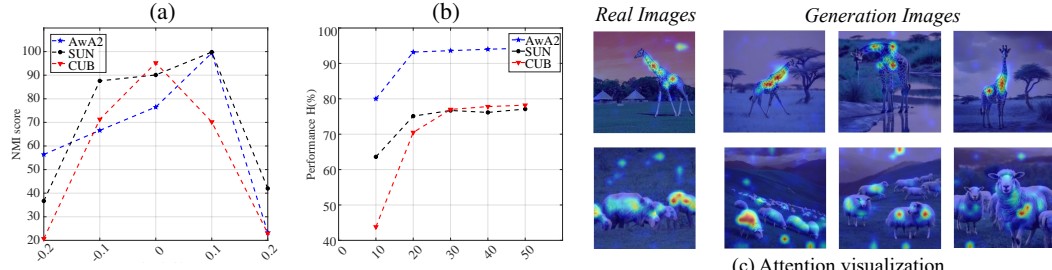

Figure 5: (a) The relationship between NMI score and T. (b) Effects of the generation numbers. (c) Attention Visualization Chefer et al. (2021) for real and generated images.

Table 3: We report the performances (%) of the models trained from AwA2 dataset and evaluated on the CUB datasets. The test space represents the predicted class number, where AwA2 and CUB comprise 50 and 200 classes, respectively. Notably, 200+50 represents that we test each sample in an independent space. FT is the abbreviation of Fine-Tuning.

| Methods | Test Space (Class Num) | AwA2 → CUB | | |
| --- | --- | --- | --- | --- |
| | | $Acc_{original}$ | $Acc_{target}$ | $Acc_H$ |
| Frozen CLIP | 250 | 89.2 | 49.6 | 63.8 |
| FT on AwA2 | 250 | 93.6 | 9.4 | 17.1 |
| Frozen CLIP | 200+50 | 92.5 | 55.0 | 68.9 |
| FT on AwA2 + HDG (Ours) | 250 | 94.1 | 58.4 | 72.1 |

**Visualizations.** To intuitively explain the effectiveness of the hierarchical grouping strategy, we randomly select 20 samples for each class to visualize the distribution by t-SNE Van der Maaten & Hinton (2008) in Fig. 4(b-c). In Fig. 4(b), we visualize the test samples from both base and novel classes with the CLIP encoder fine-tuned with the base class samples. In Fig. 4(c), we initially employ the HG strategy to partition the test samples into two distinct subsets and then visualize the feature embeddings from both subsets extracted with separate models fine-tuned with the samples in the subsets, respectively. From the results, the novel samples are clustered together with base classes in Fig. 4(b), while the distance between novel and base classes is significantly pushed away in Fig. 4(c).

**Model Generalization across Dataset.** Table 3 presents the generalization results of the model across diverse datasets. Specifically, we trained the model using the AwA2 dataset and subsequently evaluated its performance on both AwA2 and CUB datasets. The results indicate a significant decrease in performance on the CUB dataset after fine-tuning with the AwA2 dataset, hinting that the fine-tuning process with AwA2 data may have adversely affected the original CLIP model's generalization abilities. In contrast, our approach introduces an innovative method by partitioning the label space into two distinct subspaces. By training separate models, each specifically fine-tuned with samples from their respective datasets, we significantly enhance the model's adaptability and generalization capabilities.

# 5 CONCLUSION

In this paper, we propose an innovative hierarchical divide-and-conquer grouping (HDG) paradigm to address the limitations of traditional transfer learning approaches in GZSL. Unlike existing transfer learning-based methods, our approach progressively divides the unified test space into sub-spaces by measuring multi-modal distances between test samples and references, thereby constructing several independent classification tasks. This explicit division reduces prediction bias between base and novel classes. To further mitigate feature confusion within each domain, we apply the divide-and-conquer strategy to continue grouping each class into smaller, more focused subsets. During the hierarchical division process, we incorporate an LLMs-guided description generation and concept selection strategy to compensate for the scarcity of novel samples. These diverse and synthetically generated samples enhance the effectiveness of data grouping and model fine-tuning. Comprehensive evaluations demonstrate that our proposed HDG paradigm significantly outperforms the current state-of-the-art methods in GZSL.

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

# A SUPPLEMENTARY MATERIAL

## A.1 DETAILED INFORMATION OF HIERARCHICAL GROUPING

As mentioned in Section 3, we progressively divide the unified test space into hierarchical ones. The whole grouping process is given in Algorithm. 1. In addition, we also explore the optimal depth of the proposed hierarchical grouping structure. Specifically, we not only change the number of separation with balanced way (*i.e.,* keep the same depth for base and novel branches) as shown in Table. 1 of main paper, we also evaluate the unbalanced separation in Table. 4. Intuitively, when we adopt two step *i.e.,* grouping for novel branch of AwA2, we observe a slight degradation. Therefore, we speculate that the optimal depth for each datasets or different branches is inconsistent. In fact, we can search the optimal best plan by greedy search. As we experimentally explore the performance of popular benchmarks, we set the depth of each dataset by 4 to maintain the simplicity of the method.

---

**Algorithm 1** Hierarchical Grouping Strategy

---

**Input:** $x_i \in \mathcal{X}_{base}$: Base Image Sets; $\widetilde{x_i} \in \mathcal{G}$: Novel Generation Image Sets; $x_j \in \mathcal{X}_{test}$: Test Image Sets; $\mathcal{T}$ : $Threshold$
**Output:** Fine-grained class labels: $Base1, Base2$ and $Novel1, Novel2$
1: for each $x_j$ in $\mathcal{X}_{test}$:
2: **repeat**
3:  compute multi-modal distance in Eq.2 and then compute the scores of $score^d$, $score^b$ in Eq.3 and Eq.6, respectively;
4:   **while** $j < max\ number\ of\ test\ images$ and $x_j$ has not been grouped **do**
5:    **if** $score^d \geq \mathcal{T}$ **then**
6:     assign coarse-grained stage label *Base* to $x_j$
7:     **if** $score^b \geq 0$ **then**
8:      assign fine-grained stage label *Base1* to $x_j$
9:     **else**
10:      assign fine-grained stage label *Base2* to $x_j$
11:     **end if**
12:    **else**
13:     assign coarse-grained stage label *Novel* to $x_j$
14:     **if** $score^n \geq 0$ **then**
15:      assign fine-grained stage label *Novel1* to $x_j$
16:     **else**
17:      assign fine-grained stage label *Novel2* to $x_j$
18:     **end if**
19:    **stop**

---

Table 4: Effects of the proposed grouping strategy depth. Depth=2 for Base Only denotes that we conduct two stage grouping for base branch and one stage for novel classes.

| Methods | AwA2 | | | CUB | | |
|---|---|---|---|---|---|---|
| | **B** | **N** | **H** | **B** | **N** | **H** |
| Depth=1 | 93.9 | 94.2 | 94.0 | 73.8 | 75.2 | 74.5 |
| Depth=2 for Base and Novel | 94.5 | 94.1 | 94.3 | 78.4 | 78.0 | 78.2 |
| Depth=2 for Base Only | 94.5 | 94.2 | 94.3 | 78.4 | 75.2 | 76.8 |
| Depth=2 for Novel Only | 93.9 | 94.1 | 94.0 | 73.8 | 78.0 | 76.2 |

## A.2 LLMS-GUIDED DESCRIPTIONS

**Examples for LLMs-guided Descriptions.** To enhance the diversity of the generation samples, we propose a LLMs-guided method to generate various descriptions for novel classes. Here, we provide some examples which are obtained by GPT-4 on AwA2:

**bat:**

The bat hung upside down from the cave ceiling, its wings folded neatly around its body.

The bat flew gracefully through the night sky, silhouetted against the full moon.

The bat clung to the tree trunk, its sharp claws digging into the bark.

The bat's eyes gleamed in the darkness, reflecting the light from the distant streetlamp.

The bat swooped low over the field, its keen ears detecting the flutter of insect wings.

The bat nestled among the leaves, hidden from predators by its mottled brown fur.

The bat spread its wings wide, revealing the delicate webbing between its fingers.

The bat darted through the air, executing sharp turns to avoid obstacles.

The bat emitted a series of high-pitched squeaks, using echolocation to navigate.

The bat landed softly on the window ledge, its nose twitching as it sniffed the air.

**blue whale:**

The blue whale surfaced slowly, sending a towering spout of mist into the air.

The blue whale glided gracefully through the ocean, its massive body cutting through the water with ease.

The blue whale breached, its enormous body rising out of the water before crashing back down with a thunderous splash.

The blue whale's tail fluke emerged from the sea as it prepared to dive deep into the abyss.

The blue whale swam alongside a pod of dolphins, dwarfing them with its immense size.

The blue whale's mouth opened wide, filtering vast amounts of krill through its baleen plates.

The blue whale's eye, small in comparison to its body, scanned the ocean depths.

The blue whale floated near the surface, its smooth blue-gray skin glistening in the sunlight.

The blue whale's call echoed through the water, a deep, resonant sound that could travel for miles.

The blue whale moved slowly, conserving energy as it navigated the cold, nutrient-rich waters.

**bobcat**

The bobcat crouched low in the tall grass, its eyes fixed on its prey.

The bobcat leapt gracefully onto a rock, scanning the area for any signs of movement.

The bobcat's ears twitched, picking up the faint rustle of leaves in the wind.

The bobcat padded silently through the forest, its fur blending seamlessly with the underbrush.

The bobcat snarled, revealing sharp teeth as a warning to intruders.

The bobcat's tail flicked back and forth as it prepared to pounce.

The bobcat perched on a tree branch, surveying the ground below for any potential meals.

The bobcat stretched out in a patch of sunlight, enjoying the warmth on its fur.

The bobcat's keen eyes caught the glint of a rabbit's fur in the moonlight.

The bobcat slinked through the shadows, moving with a predator's stealth and precision.

**dolphin**

The dolphin leapt out of the water, performing a graceful arc before splashing back down.

The dolphin swam alongside the boat, its dorsal fin cutting through the waves.

The dolphin's playful nature was evident as it chased after a school of fish.

The dolphin's sleek, gray body glistened in the sunlight as it rode the surf.

The dolphin communicated with its pod using a series of clicks and whistles.

The dolphin flipped its tail energetically, propelling itself through the water.

The dolphin surfaced for air, its blowhole emitting a quick burst of mist.

The dolphin interacted with swimmers, gently nudging them with its snout.

The dolphin performed acrobatics, delighting the onlookers with its agility.

The dolphin's intelligent eyes observed the divers curiously as they explored the reef.

**Examples for Generation.**

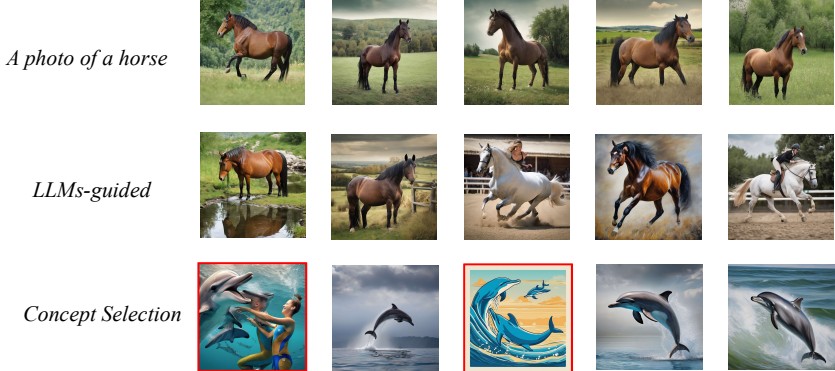

Figure 6: Examples of different descriptions.Top: The generation samples which are generated by A photo of a horse. Middle: The generation samples which are generated by LLMs-guided descriptions. Bottom: The concept selection process. The red boxes represent samples that were discarded.

In addition, we provide some examples in Fig. 6. By comparing the first row and second row, we can observe that the generation samples equipped with LLMs-guided strategy have better diversity than the others. For instance, the horses in the second row have more colors, poses and backgrounds, and are closer to the state of the horse in nature. This indicates that the proposed LLMs-guided strategy ensures diversity while not sacrificing semantic relevance. Further, we visualize the process of concept selection at the bottom of Fig. 6. We can see that the discarded samples have obviously different distribution than preserved ones.

