# OpenReview forum: "Hierarchical divide-and-conquer grouping for Zero shot learning"
_ICLR.cc/2025/Conference — ICLR 2025 Conference Withdrawn Submission_

### Official Review · Reviewer_b2fb · 2024-10-17

**Soundness:** 3
**Presentation:** 3
**Contribution:** 2
**Rating:** 5
**Confidence:** 4

**Summary:**

The paper proposes a novel framework, dubbed HDG-CLIP, for advancing Generalized Zero-Shot Learning (GZSL). To tackle the problem of prediction bias, the paper presents the hierarchical divide-and-conquer grouping (HDG) approach. Their approach divides the joint label space into base and novel subspaces and then trains a separate model for each subspace. Operating in the CLIP space (both images and the class names are features extracted from pre-trained CLIP encoders), experiments demonstrate that HDG-CLIP outperforms state-of-the-art methods by a large margin.

**Strengths:**

1. This paper proposes an interesting combination of Stable Diffusion (SD), LLM (GPT-4), and CLIP. The core idea is to divide the joint label space into base and novel subspaces and train separate models. Further, (1) To obtain visual samples of novel classes, they adopt SD to synthesize diverse visual samples; (2) To encourage the synthesized visual samples to be discriminative, they filter samples with a semantic-level concept selection strategy; (3) They leverage GPT-4 to generate class-level descriptions.
2. Experiment results are thorough and extensive - with comparison against various methods. The proposed approach outperforms other methods by a large margin on different datasets.
3. The method for improving the generalization capability of CLIP is noteworthy.

**Weaknesses:**

1. Pointing out a mistake of this paper below:

- Line 324 & 370 : GZSC ->  'GZSL'?

2. While the combination of proposed ideas is interesting and practical, the individual ideas have been used extensively in literature and are mildly novel.
3. While the authors present ablation studies on the various components they have proposed in Table 2 and Figure 4(a), they do not provide a clear baseline. For instance, it is much easier to classify in the CLIP space, but what about the performance of simply using real training images plus generated images for unseen classes to learn a classifier ?
4. The results are shown only on the three GZSL dataset. It is difficult to evaluate the model's performance in these limited experiments. I request the author use the other datasets like Flower and APY.
5. The main contribution of this paper is the hierarchical divide-and-conquer grouping paradigm. Can you show me its performance in other spaces? For example, using Resnet/ViT as the vision backbone and expert-annotated attributes to demonstrate its efficacy.  If it can still offer such a significant improvement, I would be more than happy to raise my rating.

**Questions:**

My concerns are listed in the ''Weaknesses".

---

### Official Review · Reviewer_4ND6 · 2024-10-28

**Soundness:** 2
**Presentation:** 2
**Contribution:** 2
**Rating:** 3
**Confidence:** 5

**Summary:**

This paper identifies that generalized zero-shot learning is a challenging task due to a prediction bias that the test samples are easily misclassified to the training classes. To deal with this issue, this paper proposes a divide-and-conquer strategy that segregates the label space into distinct base and novel subspaces. For the novel classes, an off-the-shelf diffusion model is used to generate novel class data samples. Additionally, each label space is further divide into two different subspace by unsupervised clustering. In testing stage, the test samples are first divided into a smaller subspace and classified by the classifier trained within the subspace.

**Strengths:**

- The paper proposes a new divide-and-conquer classification strategy for ZSL
- Together with the powerful diffusion model and GPT-4 model, the proposed ZSL method achieves the state-of-the-art performance on commonly used ZSL datasets.

**Weaknesses:**

- The proposed method primarily consists of two strategies for Zero-Shot Learning (ZSL): generating novel class images using a diffusion model and GPT-4, and proposing a divide-and-conquer classification strategy. However, the experimental results may be misleading, as it's unclear how the divide-and-conquer classification strategy would perform when applied to traditional generative methods, without the use of a diffusion model and GPT-4 for novel class generation. In other words, it is unclear where the good performance come from, the powerful diffusion model and GPT-4 model, or the proposed divide-and-conquer strategy.
- Based on the experimental results in Table 1, the performance gain appears to stem primarily from model fine-tuning. Without fine-tuning, the performance on CUB and SUN datasets is notably poor. Further explanations are needed to address this discrepancy.
- It is unclear why setting the depth of the segmented label space to 2 yields the best performance. Intuitively, the optimal depth value should depend on the number of classes in both base and novel categories. For datasets with a larger number of classes, more segmentations might be beneficial, while fewer may suffice for smaller datasets. Therefore, applying the same depth value across different datasets may not yield optimal results.
- It is hard to clearly distinguish the difference between Figure 4(b) and Figure 4(c).

**Questions:**

- Where does the good performance come from, the powerful diffusion model and GPT-4 model, or the proposed divide-and-conquer strategy?
- Why is the performance so poor without model fine-tuning?
- Why setting the depth of the segmented label space to 2 yields the best performance?

---

### Official Review · Reviewer_SSTt · 2024-10-28

**Soundness:** 2
**Presentation:** 3
**Contribution:** 1
**Rating:** 5
**Confidence:** 5

**Summary:**

This paper uses large-scale pre-trained CLIP (a vision-language model), GPT (a language model) and Stable Diffusion (a image generation model) to classify the test dataset of existing popurlar zero-shot learning (ZSL) benchmarks.
Specifically, it divides the whole class space to seen class space and unseen class space, then continually divide them to four subspace (two for the seen and anther two for the unseen).
Every subspaces are unsupervised clustered by text embedding provided by CLIP.

**Strengths:**

Clear presentations.

**Weaknesses:**

Major problem:
1. The definition of Zero-shot Learning (ZSL) is not correctly used. The authors mentioned ZSL requires the model to recognize unseen classes without additional training data, but in your method, you use pretrained CLIP and Stable Diffusion trained by large-scale dataset. For example, CLIP is trained from millions text-image pairs from web.
How can you ensure the test data is not never seen?

2. Besides, in the original paper of CLIP, it has clearly claimed they changed the definition of ZSL from class-level to dataset-level.
_"In computer vision, zero-shot learning usually refers to the study of generalizing to unseen object categories in image classification (Lampert et al., 2009). We instead use the term in a broader sense and study generalization to unseen datasets."_
﻿ This excerpt is from Section 3.2 of the CLIP paper.
There is a significant gap between CLIP’s definition of zero-shot learning and the task you are addressing, which still focuses on classifying unseen categories. This difference is substantial and cannot be overlooked.

3. In a short, I suggest the proposed method is a transductive classification method insteat of ZSL method actually since the pre-trained Stable Diffusion and CLIP models have been exposed to numerous unseen class samples.
Thus, the experiments are extramely unfair for other SOTAs.

4. The generated image and description examples about SUN are missing.

Minor problems:

5. The method types at Table1 are not correct. The Calibration-based methods still are belong to embedding methods.

6.  The method I2MVFormer is cited two times at Table1. And at the second time, the authors mis-spell it as I2VMFormer.

7. The method ICCE misses its publish time at Table1.

8. Why only divide two subspaces for seen/unseen spaces?

**Questions:**

See weakness

**Details Of Ethics Concerns:**

The paper involoves generating images from Stable Diffusion. The bias problem should be considered.

---

### Official Review · Reviewer_VJPg · 2024-11-06

**Soundness:** 3
**Presentation:** 2
**Contribution:** 2
**Rating:** 5
**Confidence:** 5

**Summary:**

This paper proposes utilizing a large language model (LLM) to generate distinct descriptions for each class, which are then used to prompt a diffusion model to synthesize examples for unseen classes. In the final prediction phase, the approach involves comparing test images against generated images for unseen classes, real images from seen classes, and the generated text descriptions. To facilitate this comparison, the paper introduces a DIVIDE-AND-CONQUER and HIERARCHICAL GROUPING strategy, designed to improve the matching accuracy between test images and generated examples across different classes.

**Strengths:**

1. The paper is well-structured and clearly written, making it easy to follow and understand.

2. The approach of using an LLM and a diffusion model to generate missing data for unseen classes is novel.

3. By generating images and descriptions, the method reframes the zero-shot learning (ZSL) problem as a retrieval-based task. The retrieval uses generated unseen images, existing seen images, and generated text descriptions, which eliminates the need for additional training.

**Weaknesses:**

1. Given the use of large models on relatively small ZSL datasets, it would be beneficial for the authors to discuss GPU utilization and the time required for generation and inference.

2. The connection between Section 3.1 and Section 3.2 appears weak. It seems that Section 3.1 primarily addresses data generation for unseen classes. A concern is whether lighter-weight generative models could achieve similar results. Comparing this approach to other image and language models could strengthen the evaluation.

3. Most importantly, how can data leakage be prevented? It is crucial to ensure that the diffusion model and the CLIP model have never been exposed to unseen test images. This issue needs to be explicitly addressed to confirm the integrity of the ZSL task.

4. To enhance clarity, it would be helpful to provide a direct comparison between real unseen images and the generated unseen images, highlighting the differences.

5. More detail is needed on the use of the Normalized Mutual Information (NMI) metric, particularly regarding how it is applied to determine thresholds in this context.

6. Regarding comparison fairness, since many existing methods do not use CLIP as the image encoder, it would be useful to conduct additional experiments with other visual backbones. This would provide a more comprehensive comparison across methods.

**Questions:**

Please see the `Weaknesses’ above.

---

### Note · Authors · 2024-11-13

I have read and agree with the venue's withdrawal policy on behalf of myself and my co-authors.